# IMPROVING SEQUENCE-TO-SEQUENCE LEARNING VIA OPTIMAL TRANSPORT

**Liqun Chen[1], Yizhe Zhang[2], Ruiyi Zhang[1], Chenyang Tao[1], Zhe Gan[3], Haichao Zhang[4], Bai Li[1], Dinghan Shen[1], Changyou Chen[5], Lawrence Carin[1]**
[1]Duke University, [2]Microsoft Research, [3]Microsoft Dynamics 365 AI Research
[4]Baidu Research, [5]SUNY at Buffalo
{liqun.chen}@duke.edu

## ABSTRACT

Sequence-to-sequence models are commonly trained via maximum likelihood estimation (MLE). However, standard MLE training considers a word-level objective, predicting the next word given the previous ground-truth partial sentence. This procedure focuses on modeling local syntactic patterns, and may fail to capture long-range semantic structure. We present a novel solution to alleviate these issues. Our approach imposes global sequence-level guidance via new supervision based on optimal transport, enabling the overall characterization and preservation of semantic features. We further show that this method can be understood as a Wasserstein gradient flow trying to match our model to the ground truth sequence distribution. Extensive experiments are conducted to validate the utility of the proposed approach, showing consistent improvements over a wide variety of NLP tasks, including machine translation, abstractive text summarization, and image captioning.

## 1 INTRODUCTION

Sequence-to-sequence (Seq2Seq) models are widely used in various natural language processing tasks, such as machine translation (Bahdanau et al., 2015; Cho et al., 2014; Sutskever et al., 2014), text summarization (Chopra et al., 2016; Rush et al., 2015) and image captioning (Vinyals et al., 2015; Xu et al., 2015). Typically, Seq2Seq models are based on an encoder-decoder architecture, with an encoder mapping a source sequence into a latent vector, and a decoder translating the latent vector into a target sequence. The goal of a Seq2Seq model is to optimize this encoder-decoder network to generate sequences close to the target. Therefore, a proper measure of the distance between sequences is crucial for model training.

Maximum likelihood estimation (MLE) is often used as the training paradigm in existing Seq2Seq models (Goodfellow et al., 2016; Lamb et al., 2016). The MLE-based approach maximizes the likelihood of the next word conditioned on its previous ground-truth words. Such an approach adopts cross-entropy loss as the objective, essentially measuring the word difference at each position of the target sequence (assuming truth for the preceding words). That is, MLE only provides a word-level training loss (Ranzato et al., 2016). Consequently, MLE-based methods suffer from the so-called exposure bias problem (Bengio et al., 2015; Ranzato et al., 2016), *i.e.*, the discrepancy between training and inference stages. During inference, each word is generated sequentially based on previously generated words. However, ground-truth words are used in each timestep during training (Huszr, 2015; Wiseman & Rush, 2016). Such discrepancy in training and testing leads to accumulated errors along the sequence-generation trajectory, and may therefore produce unstable results in practice. Further, commonly used metrics for evaluating the generated sentences at test time are sequence-level, such as BLEU (Papineni et al., 2002) and ROUGE (Lin, 2004). This also indicates a mismatch of the training loss and test-time evaluation metrics.

Attempts have been made to alleviate the above issues, via a sequence-level training loss that enables comparisons between the *entire* generated and reference sequences. Such efforts roughly fall into two categories: ($i$) reinforcement-learning-based (RL) methods (Bahdanau et al., 2017; Ranzato et al., 2016) and ($ii$) adversarial-learning-based methods (Yu et al., 2017; Zhang et al., 2017). These methods overcome the exposure bias issue through criticizing model output during training; however, both schemes have their own vulnerabilities. RL methods often suffer from large variance

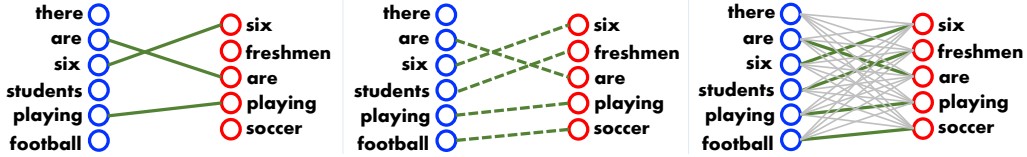

Figure 1: Different matching schemes. Left to right: hard matching, soft bipartite matching and OT matching. Dominant edges are shown in dark green for OT matching.

on policy-gradient estimation, and control variates and carefully designed baselines (such as a self-critic) are needed to make RL training more robust (Liu et al., 2018; Rennie et al., 2017). Further, the rewards used by RL training are often criticized as a bad proxy for human evaluation, as they are usually highly biased towards certain particular aspects (Wang et al., 2018b). On the other hand, adversarial supervision relies on the delicate balance of a mini-max game, which can be easily undermined by mode-trapping and gradient-vanishing problems (Arjovsky et al., 2017; Zhang et al., 2017). Sophisticated tuning is often desired for successful adversarial training.

We present a novel Seq2Seq learning scheme that leverages *optimal transport* (OT) to construct sequence-level loss. Specifically, the OT objective aims to find an optimal matching of similar words/phrases between two sequences, providing a way to promote their semantic similarity (Kusner et al., 2015). Compared with the above RL and adversarial schemes, our approach has: ($i$) semantic-invariance, allowing better preservation of sequence-level semantic information; and ($ii$) improved robustness, since neither the reinforce gradient nor the mini-max game is involved. The OT loss allows end-to-end supervised training and acts as an effective sequence-level regularization to the MLE loss.

Another novel strategy distinguishing our model from previous approaches is that during training we consider not only the OT distance between the generated sentence and ground-truth references, but also the OT distance between the generated sentence and its corresponding input. This enables our model to simultaneously match the generated output sentence with both the source sentence(s) and target reference sentence, thus enforcing the generator to leverage information contained in the input sentence(s) during generation.

The main contributions of this paper are summarized as follows. ($i$) A new sequence-level training algorithm based on optimal transport is proposed for Seq2Seq learning. In practice, the OT distance is introduced as a regularization term to the MLE training loss. ($ii$) Our model can be interpreted as approximate Wasserstein gradient flows, learning to approximately match the sequence distribution induced by the generator and a target data distribution. ($iii$) In order to demonstrate the versatility of the proposed method, we conduct extensive empirical evaluations on three tasks: machine translation, text summarization, and image captioning.

## 2 SEMANTIC MATCHING WITH OPTIMAL TRANSPORT

We consider two components of a sentence: its syntactic and semantic parts. In a Seq2Seq model, it is often desirable to keep the semantic meaning while the syntactic part can be more flexible. Conventional training schemes, such as MLE, are known to be well-suited for capturing the syntactic structure. As such, we focus on the semantic part. An intuitive way to assess semantic similarity is to directly match the "key words" between the synthesized and the reference sequences. Consider the respective sequences as sets $\mathbb{A}$ and $\mathbb{B}$, with vocabularies as their elements. Then the matching can be evaluated by $|\mathbb{A} \cap \mathbb{B}|$, where $|\cdot|$ is the counting measure for sets. We call this *hard matching*, as it seeks to exactly match words from both sequences.

For language models, the above hard matching could be an over simplification. This is because words have semantic meaning, and two different words can be close to each other in the semantic space. To account for such ambiguity, we can relax the hard matching to *soft bipartite matching* (SBM). More specifically, assuming all sequences have the same length $n$, we pair $\boldsymbol{w}_{i_k} \in \mathbb{A}$ and $\boldsymbol{w}'_{j_k} \in \mathbb{B}$ for $k \in [1, K]$, such that $K \leq n$, $\{i_k\}, \{j_k\}$ are unique and $\mathcal{L}_{\text{SBM}} = \sum_k c(\boldsymbol{w}_{i_k}, \boldsymbol{w}'_{j_k})$ is minimized. Here $c(\boldsymbol{w}, \boldsymbol{w}')$ is a cost function measuring the semantic dissimilarity between the two words. For instance, the cosine distance $c(\boldsymbol{x}, \boldsymbol{y}) = 1 - \frac{\boldsymbol{x}^\top \boldsymbol{y}}{\|\boldsymbol{x}\|_2 \|\boldsymbol{y}\|_2}$ between two word embedding vectors $\boldsymbol{x}$ and $\boldsymbol{y}$ is a popular choice (Pennington et al., 2014). This minimization can be solved exactly,

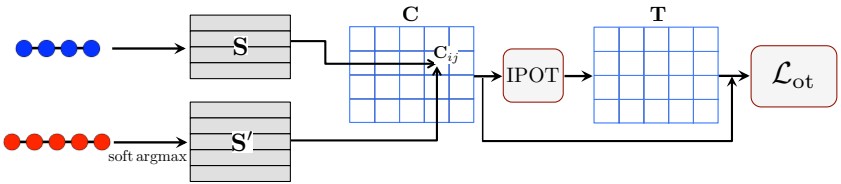

Figure 2: Schematic computation graph of OT loss.

*e.g.*, via the Hungarian algorithm (Kuhn, 1955). Unfortunately, its $O(n^3)$ complexity scales badly for common NLP tasks, and the objective is also non-differentiable wrt model parameters. As such, end-to-end supervised training is not feasible with the Hungarian matching scheme. To overcome this difficulty, we propose to further relax the matching criteria while keeping the favorable features of a semantic bipartite matching. OT arises as a natural candidate.

## 2.1 OPTIMAL TRANSPORT AND WASSERSTEIN DISTANCE

We first provide a brief review of optimal transport, which defines distances between probability measures on a domain $\mathbb{X}$ (the sequence space in our setting). The *optimal transport distance* for two probability measures $\mu$ and $\nu$ is defined as (Peyré et al., 2017):

$$\mathcal{D}_c(\mu, \nu) = \inf_{\gamma \in \Pi(\mu, \nu)} \mathbb{E}_{(\boldsymbol{x}, \boldsymbol{y}) \sim \gamma} \left[ c(\boldsymbol{x}, \boldsymbol{y}) \right], \tag{1}$$

where $\Pi(\mu, \nu)$ denotes the set of all joint distributions $\gamma(\boldsymbol{x}, \boldsymbol{y})$ with marginals $\mu(\boldsymbol{x})$ and $\nu(\boldsymbol{y})$; $c(\boldsymbol{x}, \boldsymbol{y}) : \mathbb{X} \times \mathbb{X} \to \mathbb{R}$ is the cost function for moving $\boldsymbol{x}$ to $\boldsymbol{y}$, *e.g.*, the Euclidean or cosine distance. Intuitively, the optimal transport distance is the minimum cost that $\gamma$ induces in order to transport from $\mu$ to $\nu$. When $c(\boldsymbol{x}, \boldsymbol{y})$ is a metric on $\mathbb{X}$, $\mathcal{D}_c(\mu, \nu)$ induces a proper metric on the space of probability distributions supported on $\mathbb{X}$, commonly known as the Wasserstein distance (Villani, 2008). One of the most popular choices is the $2-$Wasserstein distance $W_2^2(\mu, \nu)$ where the squared Euclidean distance $c(\boldsymbol{x}, \boldsymbol{y}) = \|\boldsymbol{x} - \boldsymbol{y}\|^2$ is used as cost.

**OT distance on discrete domains** We mainly focus on applying the OT distance on textual data. Therefore, we only consider OT between discrete distributions. Specifically, consider two discrete distributions $\boldsymbol{\mu}, \boldsymbol{\nu} \in \mathbf{P}(\mathbb{X})$, which can be written as $\boldsymbol{\mu} = \sum_{i=1}^n u_i \delta_{\mathbf{x}_i}$ and $\boldsymbol{\nu} = \sum_{j=1}^m v_j \delta_{\mathbf{y}_j}$ with $\delta_{\mathbf{x}}$ the Dirac function centered on $\mathbf{x}$. The weight vectors $\mathbf{u} = \{u_i\}_{i=1}^n \in \Delta_n$ and $\mathbf{v} = \{v_i\}_{i=1}^m \in \Delta_m$ respectively belong to the $n$ and $m$-dimensional simplex, *i.e.*, $\sum_{i=1}^n u_i = \sum_{j=1}^m v_j = 1$, as both $\boldsymbol{\mu}$ and $\boldsymbol{\nu}$ are probability distributions. Under such a setting, computing the OT distance as defined in (1) is equivalent to solving the following network-flow problem (Luise et al., 2018):

$$\mathcal{L}_{\text{ot}}(\boldsymbol{\mu}, \boldsymbol{\nu}) = \min_{\mathbf{T} \in \Pi(\mathbf{u}, \mathbf{v})} \sum_{i=1}^n \sum_{j=1}^m \mathbf{T}_{ij} \cdot c(\boldsymbol{x}_i, \boldsymbol{y}_j) = \min_{\mathbf{T} \in \Pi(\mathbf{u}, \mathbf{v})} \langle \mathbf{T}, \mathbf{C} \rangle, \tag{2}$$

where $\Pi(\mathbf{u}, \mathbf{v}) = \{\mathbf{T} \in \mathbb{R}_+^{n \times m} | \mathbf{T} \mathbf{1}_m = \mathbf{u}, \mathbf{T}^\top \mathbf{1}_n = \mathbf{v}\}$, $\mathbf{1}_n$ denotes an $n$-dimensional all-one vector, $\mathbf{C}$ is the cost matrix given by $\mathbf{C}_{ij} = c(\boldsymbol{x}_i, \boldsymbol{y}_j)$ and $\langle \mathbf{T}, \mathbf{C} \rangle = \text{Tr}(\mathbf{T}^\top \mathbf{C})$ represents the Frobenius dot-product. We refer to the minimizer $\mathbf{T}^*$ of (2) as *OT matching*. Comparing the two objectives, one can readily recognize that soft bipartite matching represents a special constrained solution to (2), where $\mathbf{T}$ can only take values in $\Gamma = \{\mathbf{T} | \max_i \{\|\mathbf{T} \mathbf{e}_i\|_0, \|\mathbf{e}_i^T \mathbf{T}\|_0\} \leq 1, \mathbf{T}_{ij} \in \{0, 1\}, \|\mathbf{T}\|_0 = K\}$ instead of $\Pi(\mathbf{u}, \mathbf{v})$; here $\|\cdot\|_0$ is the $L_0$ norm and $\mathbf{e}_i$ is the unit vector along $i$-th axis. As such, OT matching can be regarded as a relaxed version of soft bipartite matching. In Figure 1 we illustrate the three matching schemes discussed above.

**The IPOT algorithm** Unfortunately, the exact minimization over $\mathbf{T}$ is in general computational intractable (Arjovsky et al., 2017; Genevay et al., 2018; Salimans et al., 2018). To overcome such intractability, we consider an efficient iterative approach to approximate the OT distance. We propose to use the recently introduced Inexact Proximal point method for Optimal Transport (IPOT) algorithm to compute the OT matrix $\mathbf{T}^*$, thus also the OT distance (Xie et al., 2018). IPOT provides a solution to the original OT problem specified in (2). Specifically, IPOT iteratively solves the following optimization problem using the proximal point method (Boyd & Vandenberghe, 2004):

$$\mathbf{T}^{(t+1)} = \underset{\mathbf{T} \in \Pi(\boldsymbol{x}, \boldsymbol{y})}{\arg \min} \left\{ \langle \mathbf{T}, \mathbf{C} \rangle + \beta \cdot \mathcal{B}(\mathbf{T}, \mathbf{T}^{(t)}) \right\}, \tag{3}$$

where the proximity metric term $\mathcal{B}(\mathbf{T}, \mathbf{T}^{(t)})$ penalizes solutions that are too distant from the latest approximation, and $\frac{1}{\beta}$ is understood as the generalized stepsize. This renders a tractable iterative scheme towards the exact OT solution. In this work, we employ the generalized KL Bregman divergence $\mathcal{B}(\mathbf{T}, \mathbf{T}^{(t)}) = \sum_{i,j} \mathbf{T}_{ij} \log \frac{\mathbf{T}_{ij}}{\mathbf{T}_{ij}^{(t)}} - \sum_{i,j} \mathbf{T}_{ij} + \sum_{i,j} \mathbf{T}_{ij}^{(t)}$ as the proximity metric. Algorithm

1 describes the implementation details for IPOT.
Note that the Sinkhorn algorithm (Cuturi, 2013) can also be used to compute the OT matrix. Specifically, the Sinkhorn algorithm tries to solve the entropy regularized optimization problem: $\hat{\mathcal{L}}_{\text{ot}}(\boldsymbol{\mu}, \boldsymbol{\nu}) = \min_{\mathbf{T} \in \Pi(\mathbf{u}, \mathbf{v})} \langle \mathbf{T}, \mathbf{C} \rangle - \frac{1}{\epsilon} H(\mathbf{T})$, where $H(\mathbf{T}) = -\sum_{i,j} \mathbf{T}_{ij}(\log(\mathbf{T}_{ij}) - 1)$ is the entropy regularization term and $\epsilon > 0$ is the regularization strength. However, in our experiments, we empirically found that the numerical stability and performance of the Sinkhorn algorithm is quite sensitive to the choice of the hyper-parameter $\epsilon$, thus only IPOT is considered in our model training.

---

**Algorithm 1** IPOT algorithm

---

1: **Input:** Feature vectors $\mathbf{S} = \{\mathbf{z}_i\}_1^n, \mathbf{S}' = \{\mathbf{z}'_j\}_1^m$ and generalized stepsize $1/\beta$,
2: $\boldsymbol{\sigma} = \frac{1}{m}\mathbf{1_m}, \mathbf{T}^{(1)} = \mathbf{1_n}\mathbf{1_m}^\top$
3: $\mathbf{C}_{ij} = c(\mathbf{z}_i, \mathbf{z}'_j), \mathbf{A}_{ij} = e^{-\frac{\mathbf{C}_{ij}}{\beta}}$
4: **for** $t = 1, 2, 3 \ldots$ **do**
5: $\quad \mathbf{Q} = \mathbf{A} \odot \mathbf{T}^{(t)}$ // $\odot$ is Hadamard product
6: $\quad$ **for** $k = 1, \ldots K$ **do** // $K = 1$ in practice
7: $\quad\quad \boldsymbol{\delta} = \frac{1}{n\mathbf{Q}\boldsymbol{\sigma}}, \boldsymbol{\sigma} = \frac{1}{m\mathbf{Q}^\top\boldsymbol{\delta}}$
8: $\quad$ **end for**
9: $\quad \mathbf{T}^{(t+1)} = \text{diag}(\boldsymbol{\delta})\mathbf{Q}\text{diag}(\boldsymbol{\sigma})$
10: **end for**
11: **Return** $\langle \mathbf{T}, \mathbf{C} \rangle$

---

## 2.2 OPTIMAL TRANSPORT DISTANCE AS A SEQUENCE LEVEL LOSS

Figure 2 illustrates how OT is computed to construct the sequence-level loss. Given two sentences, we can construct their word-level or phrase-level embedding matrices $\mathbf{S}$ and $\mathbf{S}'$, where $\mathbf{S} = \{\mathbf{z}_i\}$ is usually recognized as the reference sequence embedding and $\mathbf{S}' = \{\mathbf{z}'_j\}$ for the model output sequence embedding. The cost matrix $\mathbf{C}$ is then computed by $\mathbf{C}_{ij} = c(\mathbf{z}_i, \mathbf{z}'_j)$ and passed on to the IPOT algorithm to get the OT distance. Our full algorithm is summarized in Algorithm 2, and more detailed model specifications are given below.

**Encoding model belief with a differentiable sequence generator** We first describe how to design a differentiable sequence generator so that the gradients can be backpropagated from the OT losses to update the model belief. The Long Short-Term Memory (LSTM) recurrent neural network (Hochreiter & Schmidhuber, 1997) is used as our sequence model. At each timestep $t$, the LSTM decoder outputs a logit vector $\boldsymbol{v}_t$ for the vocabularies, based on its context. Directly sampling from the multinomial distribution $\hat{\boldsymbol{w}}_t \sim \text{Softmax}(\boldsymbol{v}_t)$ is a non-differentiable operation[1], so we consider the following differentiable alternatives:

- *Soft-argmax*: $\hat{\boldsymbol{w}}_t^{SA} = \text{Softmax}(\boldsymbol{v}_t/\tau)$, where $\tau \in (0, 1)$ is the annealing parameter (Zhang et al., 2017). This approximates the deterministic sampling scheme $\hat{\boldsymbol{w}}_t^{\max} = \arg\max\{\boldsymbol{v}_t\}$;
- *Gumbel-softmax* (GS): $\hat{\boldsymbol{w}}_t^{GS} = \text{Softmax}((\boldsymbol{v}_t + \boldsymbol{\xi}_t)/\tau)$, where $\boldsymbol{\xi}_t$ are iid Gumbel random variables for each of the vocabulary. It is also known as the Concrete distribution (Jang et al., 2016; Maddison et al., 2017).

Unstable training and sub-optimal solutions have been observed for the GS-based scheme for the Seq2Seq tasks we considered (see Appendix G, Table 11), possibly due to the extra uncertainty introduced. As such, we will assume the use of soft-argmax to encode model belief in $\hat{\boldsymbol{w}}_t$ unless otherwise specified. Note $\hat{\boldsymbol{w}}_t$ is a normalized non-negative vector that sums up to one.

**Sequence-level OT-matching loss** To pass on the model belief to the OT loss, we use the mean word embedding predicted by the model, given by $\hat{\boldsymbol{z}}_t = \mathbf{E}^T \hat{\boldsymbol{w}}_t$, where $\mathbf{E} \in \mathbb{R}^{V \times d}$ is the word embedding matrix, $V$ is the vocabulary size and $d$ is the dimension for the embedding vector. We collect the predicted sequence embeddings into $\mathbf{S}_g = \{\hat{\boldsymbol{z}}_t\}_{t=1}^L$, where $L$ is the length of sequence. Similarly we denote the reference sequence embeddings as $\mathbf{S}_r = \{\boldsymbol{z}_t\}_{t=1}^L$, using ground truth one-hot input token sequence $\{\boldsymbol{w}_t\}$. Based on the sequence embeddings $\mathbf{S}_r$ and $\mathbf{S}_g$, we can compute the sequence-level OT loss between ground-truth and model prediction using the IPOT algorithm described above for different Seq2Seq tasks:

$$\mathcal{L}_{\text{seq}} \triangleq \text{IPOT}(\mathbf{S}_g, \mathbf{S}_r). \tag{4}$$

---

[1]Here $\hat{\boldsymbol{w}}_t$ is understood as an one-hot vector in order to be notationally consistent with its differentiable alternatives.

---

**Algorithm 2** Seq2Seq Learning via Optimal Transport.

---

1: **Input:** batch size $m$, paired input and output sequences $(\mathbf{X}, \mathbf{Y})$
2: Load MLE pre-trained Seq2Seq model $\mathcal{M}(\cdot; \theta)$ and word embedding $\mathbf{E}$
3: **for** iteration $= 1, \ldots$ MaxIter **do**
4:     **for** $i = 1, \ldots, m$ **do**
5:         Draw a pair of sequences $\boldsymbol{x}_i, \boldsymbol{y}_i \sim (\mathbf{X}, \mathbf{Y})$, where $\boldsymbol{x}_i = \{\tilde{\boldsymbol{w}}_{i,t}\}, \boldsymbol{y}_i = \{\boldsymbol{w}_{i,t}\}$
6:         Compute logit vectors from model: $\{\boldsymbol{v}_{i,t}\} = \mathcal{M}(\boldsymbol{x}_i; \theta)$
7:         Encode model belief: $\hat{\boldsymbol{w}}_{i,t} = \text{Soft-argmax}(\boldsymbol{v}_{i,t})$
8:         Feature vector embedding: $\mathbf{S}_{r,i} = \{\mathbf{E}^T \boldsymbol{w}_{i,t}\}, \mathbf{S}_{g,i} = \{\mathbf{E}^T \hat{\boldsymbol{w}}_{i,t}\}$
9:     **end for**
10:    Update the $\mathcal{M}(\cdot; \theta)$ by optimizing: $\frac{1}{m} \sum_{i=1}^{m} [\mathcal{L}_{\text{MLE}}(\boldsymbol{x}_i, \boldsymbol{y}_i; \theta) + \gamma \mathcal{L}_{\text{seq}}(\mathbf{S}_{r,i}, \mathbf{S}_{g,i})]$
11: **end for**

---

**Soft-copying mechanism** We additionally consider feature matching using the OT criteria between the source and target. Intuitively, it will encourage the global semantic meaning to be preserved from source to target. This is related to the copy network (Gu et al., 2016). However, in our framework, the copying mechanism can be understood as a soft optimal-transport-based copying, instead of the original hard retrieved-based copying used by Gu et al. (2016). This soft copying mechanism considers semantic similarity in the embedding space, and thus presumably delivers smoother transformation of information. In the case where the source and target sequences do not share vocabulary (*e.g.*, machine translation), this objective can still be applied by sharing the word embedding space between source and target. Ideally, the embedding for the same concept in different languages will automatically be aligned by optimizing such loss, making available a cosine-similarity-based cost matrix. This is also related to bilingual skip-gram (Luong et al., 2015b). We denote this loss as $\mathcal{L}_{\text{copy}} \triangleq \text{IPOT}(\mathbf{S}_g, \mathbf{S}_s)$, where $\mathbf{S}_s$ represents the source sequence embeddings.

**Complementing MLE training with OT regularization** OT training objectives discussed above can not train a proper language model on its own, as they do not explicitly consider word ordering, *i.e.*, the syntactic strucuture of a language model. To overcome this issue, we propose to combine the OT loss with the *de facto* likelihood loss $\mathcal{L}_{\text{MLE}}$, which gives us the final training objective: $\mathcal{L} = \mathcal{L}_{\text{MLE}} + \gamma \mathcal{L}_{\text{seq}}$, where $\gamma > 0$ is a hyper-parameter to be tuned. For tasks with both input and output sentences, such as machine translation and text summarization, $\mathcal{L}_{\text{copy}}$ can be applied, in which case the final objective can be written as $\mathcal{L} = \mathcal{L}_{\text{MLE}} + \gamma_1 \mathcal{L}_{\text{copy}} + \gamma_2 \mathcal{L}_{\text{seq}}$.

## 3    Interpretation as Approximate Wasserstein Gradient Flows

To further justify the use of our approach (minimizing the loss $\{\mathcal{L}_{\text{MLE}} + \gamma \mathcal{L}_{ot}\}$, where $\mathcal{L}_{ot}$ denotes the Wasserstein loss), we now explain how our model approximately learns to match the ground-truth sequence distribution. Our derivation is based on the theory of Wasserstein gradient flows (WGF) (Villani, 2008). In WGF, the Wasserstein distance describes the local geometry of a trajectory in the space of probability measures converging to a target distribution (Ambrosio et al., 2005). In the following, we show that the proposed method learns to approximately match the data distribution, from the perspective of WGF. For simplicity we only discuss the continuous case, while a similar argument also holds for the discrete case (Li & Montufar, 2018).

We denote the induced distribution of the sequences generated from the decoder at the $l$-th iteration as $\mu_l$. Assume the sequence data distribution is given by $p_d(\mathbf{x})$. Intuitively, the optimal generator in a Seq2Seq model learns a distribution $\mu^*(\mathbf{x})$ that matches $p_d(\mathbf{x})$. Based on Craig (2014), this can be achieved by composing a sequence of discretized WGFs given by:

$$\mu_l = J_h(\mu_{l-1}) = J_h(J_h(\cdots(\mu_0))) \,, \tag{5}$$

with $J_h(\cdot)$ defined as

$$J_h(\mu) = \underset{\nu \in \mathcal{P}_s}{\arg\min} \left\{ \frac{1}{2h} W_2^2(\mu, \nu) + D_{\text{KL}}(\nu \| p_d) \right\} = \underset{\nu \in \mathcal{P}_s}{\arg\min} \{\mathcal{L}_{\text{WGF}}(\mu, \nu)\} \,, \tag{6}$$

where $\lambda = 1/(2h)$ is a regularization parameter ($h$ is the generalized learning rate); $W_2^2(\mu, \nu)$ denotes the 2-Wasserstein distance between $\mu$ and $\nu$; $\mathcal{P}_s$ is the space of distributions with finite 2nd-order moments; and $D_{\text{KL}}(\nu \| p_d) = \mathbb{E}_{\mathbf{x} \sim \nu}[\log \nu(\mathbf{x}) - \log p_d(\mathbf{x})]$ is the Kullback-Leibler (KL)

divergence. It is not difficult to see that discreteized WGF is essentially optimizing the KL divergence with a proximal descent scheme, using the 2-Wasserstein distance as the proximity metric.

We denote $\mu_h^* = \lim_{l \to \infty} \mu_l$ with generalized learning rate $h$. It is well known that $\lim_{h \to 0} \mu_h^* = p_d$ (Chen et al., 2018a), that is to say the induced model distribution $\mu_l$ asymptotically converges to the data distribution $p_d$. In our case, instead of using $\mathcal{L}_{\text{WGF}}(\mu, \nu)$ as the loss function, we define a surrogate loss using its upper bound $\mathcal{L}_{\text{WGF}}(\mu, \nu) \leq \mathcal{L}_{\text{WGF}}(p_d, \nu)$, where the inequality holds because (6) converges to $p_d$. When our model distribution $\mu$ is parameterized by $\theta$, $\mu_l$ can be solved with stochastic updates on $\theta$ based on the following equation with stepsize $\eta$:

$$\theta_l \leftarrow \theta_{l-1} + \eta \nabla_\theta \mathcal{L}_{\text{WGF}}(p_d, \mu_{l-1}) = \theta_{l-1} + \eta \{ \nabla_\theta D_{\text{KL}}(\mu_{l-1} \parallel p_d) + \frac{1}{2h} \nabla_\theta W_2^2(p_d, \mu_{l-1}) \} . \quad (7)$$

Unfortunately, (7) is an infeasible update as we do not know $p_d$. However, we argue that this update is still locally valid when current model approximation $\mu_{l-1}$ is close to $p_d$. To see this, recall that the KL-divergence is a natural Riemannian metric on the space of probability measures (Amari, 1985), therefore it is locally symmetric. So we can safely replace the $D_{\text{KL}}(\mu \parallel p_d)$ term with $D_{\text{KL}}(p_d \parallel \mu)$ when $\mu$ is close to $p_d$. This recovers the loss function $\mathcal{L}_{\text{MLE}} + \gamma \mathcal{L}_{\text{seq}}$ derived in Section 2.2 as $D_{\text{KL}}(p_d \parallel \mu) = \mathcal{L}_{\text{MLE}} + H(p_d)$, where $H(p_d)$ is the entropy of $p_d$, independent of $\mu$, and $\mathcal{L}_{\text{seq}} = W_2^2(p_d, \mu)$. This justifies the use of our proposed scheme in a model-refinement stage, where model distribution $\mu$ is sufficiently close to $p_d$. Empirically, we have observed that our scheme also improves training even when $\mu$ is distant from $p_d$. While the above justification is developed based on Euclidean transport, other non-Euclidean costs such as cosine distance usually yield better empirical performance as they are more adjusted to the geometry of sequence data.

## 4    RELATED WORK AND DISCUSSION

**Optimal transport in NLP**    Although widely used in other fields such as computer vision (Rubner et al., 2000), OT has only been applied in NLP recently. Pioneered by the work of Kusner et al. (2015) on *word mover's distance* (WMD), existing literature primarily considers OT either on a macroscopic level like topic modeling (Huang et al., 2016), or a microscopic level such as word embedding (Xu et al., 2018). Euclidean distance, instead of other more general distance, is often used as the transportation cost, in order to approximate the OT distance with the Kantorovich-Rubinstein duality (Gulrajani et al., 2017) or a more efficient yet less accurate lower bound (Kusner et al., 2015). Our work employs OT for mesoscopic sequence-to-sequence models, presenting an efficient IPOT-based implementation to enable end-to-end learning for general cost functions. The proposed OT not only refines the word embedding matrix but also improves the Seq2Seq model (see Appendix H for details).

**RL for sequence generation**    A commonly employed strategy for sequence-level training is via reinforcement learning (RL). Typically, this type of method employs RL by considering the evaluation metrics as the reward to guide the generation (Bahdanau et al., 2017; Huang et al., 2018; Ranzato et al., 2016; Rennie et al., 2017; Zhang et al., 2018a). However, these approaches often introduce procedures that may yield large-variance gradients, resulting in unstable training. Moreover, it has been recognized that these automatic metrics may have poor correlation with human judgments in many scenarios (Wang et al., 2018b). As such, reinforcing the evaluation metrics can potentially boost the quantitative scores but not necessarily improve the generation quality, as such metrics usually encourage exact text snippets overlapping rather than semantic similarity. Some nonstandard metrics like SPICE (Anderson et al., 2016) also consider semantic similarity, however they also can not learn a good model on their own (Liu et al., 2017). Unlike RL methods, our method requires no human-defined rewards, thus preventing the model from over-fitting to one specific metric. As a concrete example, the two semantically similar sentences *"do you want to have lunch with us "* and *"would you like to join us for lunch"* would be considered as a bad match based on automatic metrics like BLEU, however, be rated as reasonable match in OT objective.

**GAN for sequence generation**    Another type of method adopts the framework of generative adversarial networks (GANs) (Goodfellow et al., 2014), by providing sequence-level guidance based on a learned discriminator (or, critic). To construct such a loss, Fedus et al. (2018); Guo et al. (2018); Lin et al. (2017); Yu et al. (2017) combine the policy-gradient algorithm with the original GAN training procedure, while Chen et al. (2018b); Zhang et al. (2017) uses a so-called feature mover distance and maximum mean discrepancy (MMD) to match features of real and generated sentences, respectively.

Table 1: BLEU scores on VI-EN and EN-VI.  Table 2: BLEU scores on DE-EN and EN-DE.

| Systems | NT2012 | NT2013 | Systems | NT2013 | NT2015 |
|---|---|---|---|---|---|
| VI-EN: GNMT | 20.7 | 23.8 | DE-EN: GNMT | 29.0 | 29.9 |
| VI-EN: $GNMT+\mathcal{L}_{seq}$ | 21.9 | 25.4 | DE-EN: $GNMT+\mathcal{L}_{seq}$ | 29.1 | 29.9 |
| VI-EN: $GNMT+\mathcal{L}_{seq}+\mathcal{L}_{copy}$ | **21.9** | **25.5** | DE-EN: $GNMT+\mathcal{L}_{seq}+\mathcal{L}_{copy}$ | **29.2** | **30.1** |
| EN-VI: GNMT | 23.8 | 26.1 | EN-DE: GNMT | 24.3 | 26.5 |
| EN-VI: $GNMT+\mathcal{L}_{seq}$ | 24.4 | 26.5 | EN-DE: $GNMT+\mathcal{L}_{seq}$ | 24.3 | 26.6 |
| EN-VI: $GNMT+\mathcal{L}_{seq}+\mathcal{L}_{copy}$ | **24.5** | **26.9** | EN-DE: $GNMT+\mathcal{L}_{seq}+\mathcal{L}_{copy}$ | **24.6** | **26.8** |

However, mode-collapse and gradient-vanishing problems make the training of these methods challenging. Unlike GAN methods, since no min-max games are involved, the training of our model is more robust. Moreover, compared with GAN, no additional critic is introduced in our model, which makes the model complexity comparable to MLE and less demanding to tune.

## 5 EXPERIMENTS

We consider a wide range of NLP tasks to experimentally validate the proposed model, and benchmark it with other strong baselines. All experiments are implemented with Tensorflow and run on a single NVIDIA TITAN X GPU. Code for our experiments are available from `https://github.com/LiqunChen0606/Seq2Seq-OT`.

### 5.1 NEURAL MACHINE TRANSLATION

We test our model on two datasets: ($i$) a small-scale English-Vietnamese parallel corpus of TED-talks, which has 133K sentence pairs from the IWSLT Evaluation Campaign (Cettolo et al., 2015); and ($ii$) a large-scale English-German parallel corpus with 4.5M sentence pairs, from the WMT Evaluation Campaign (Vaswani et al., 2017). We used Google's Neural Machine Translation (GMNT) model (Wu et al., 2016) as our baseline, following the architecture and hyper-parameter settings from the GNMT repository[2] to make a fair comparison. For the English-Vietnamese (*i.e.*, VI-EN and EN-VI) tasks, a 2-layer LSTM with 512 units in each layer is adopted as the decoder, with a 1-layer bidirectional-LSTM adopted as the encoder; the word embedding dimension is set to 512. Attention proposed in Luong et al. (2015a) is used together with a dropout rate of $0.2$. For the English-German (*i.e.*, DE-EN and EN-DE) tasks, we train a 4-layer LSTM decoder with 1024 units in each layer. A 2-layer bidirectional-LSTM is used as the encoder, and we adopt the attention used in Wu et al. (2016). The word embedding dimension is set to 1024. Standard stochastic gradient descent is used for training with a decreasing learning rate, and we set $\beta = 0.5$ for the IPOT algorithm. More training details are provided in Appendix A. In terms of wall-clock time, our model only slightly increases training time. For the German-English task, it took roughly 5.5 days to train the GNMT model, and 6 days to train our proposed model from scratch, which only amounts to a roughly $10\%$ increase.

We apply different combinations of $\mathcal{L}_{copy}$ and $\mathcal{L}_{seq}$ to fine-tune the pre-trained GNMT model (Luong et al., 2018) and the results are summarized in Table 1 and 2. . Additional results for training from scratch are provided in Appendix B. The proposed OT approach consistently improves upon MLE training in all experimental setups.

We additionally tested our model with a more expressive 8-layer LSTM model on the EN-DE task. The BLEU score of our method is 28.0 on NT2015. For reference, the GNMT model (same architecture) and a Transformer model (Vaswani et al., 2017) respectively report a score of 27.6 and 27.3. Our method outperforms both baselines, and it is also competitive to the state-of-the-art BLEU score 28.4 reported by Vaswani et al. (2017) using a highly sophisticated model design.

The German-to-English translation examples are provided in Table 3 for qualitative assessment. The main differences among the reference translation, our translation and the GNMT translation are highlighted in blue and red. Our OT-augmented translations are more faithful to the reference than its MLE-trained counterpart. The soft-copying mechanism introduced by OT successfully maintains the key semantic content from the reference. Presumably, the OT loss helps refine the word embedding matrices, and promotes matching between words with similar semantic meanings. Vanilla

---

[2]`https://github.com/tensorflow/nmt`

Table 3: Comparison of German-to-English translation examples. Matched key phrases are shown in the same color. First example: "*May*" is not the date when the new prime minister visited Japan, but actually is the time he won the election. Second example: GNMT's paraphrase choices are not as accurate as ours. Third example: "*nominating committee*" is controlled by the government, not a "*UN-controlled nomination committee*" in GNMT's result, and it also fails to capture the word "*retain*".

| | |
|---|---|
| Reference: | India's new prime minister, Narendra Modi, is meeting his Japanese counterpart, Shinzo Abe, in Tokyo to discuss economic and security ties, on his first major foreign visit since winning May's election. |
| GNMT: | India s new prime minister , Narendra Modi , is meeting his Japanese counterpart , Shinzo Abe , in Tokyo at his first important visit abroad in May to discuss economic and security relations . |
| Ours: | India s new Prime Minister Narendra Modi meets his Japanese counterpart , Shinzo Abe , in Tokyo at his first major foreign visit since his election in May in order to discuss economic and security relations . |
| Reference: | The next day, turning up for work as usual, she was knocked down by a motorcyclist who had mounted the pavement in what passers-by described as a "vicious rage." |
| GNMT: | The next day , when she went to work as usual , she was driven by a motorcyclist who , as passants described , went on foot in a kind of "brutal anger " . |
| Ours: | The next day , when she went to work as usual , she was crossed by a motorcyclist who , was described by passers-by , in a sort of " brutal rage " on the road . |
| Reference: | Chinese leaders presented the Sunday ruling as a democratic breakthrough because it gives Hong Kongers a direct vote, but the decision also makes clear that Chinese leaders would retain a firm hold on the process through a nominating committee tightly controlled by Beijing. |
| GNMT: | The Chinese leadership presented Sunday ' s decision as a democratic breakthrough , because Hong Kong' s citizens have a direct right to vote , but the decision also makes it clear that the Chinese leadership is firmly in control of the process through a UN-controlled nomination committee . |
| Ours: | The Chinese leadership presented Sunday ' s decision as a democratic breakthrough because it gives the citizens of Hong Kong a direct right to vote , but the decision also makes it clear that the Chinese leadership keeps the process firmly in the hands of a government-controlled Nomination Committee . |

Table 4: ROUGE scores on Gigaword.

| Systems | RG-1 | RG-2 | RG-L |
|---|---|---|---|
| Seq2Seq | 33.4 | 15.7 | 32.4 |
| Seq2Seq+$\mathcal{L}_{seq}$ | 35.8 | 17.5 | 33.7 |
| Seq2Seq+$\mathcal{L}_{seq}$+$\mathcal{L}_{copy}$ | **36.2** | **18.1** | **34.0** |

Table 5: ROUGE scores on DUC2004.

| Systems | RG-1 | RG-2 | RG-L |
|---|---|---|---|
| Seq2Seq | 28.0 | 9.4 | 24.8 |
| Seq2Seq+$\mathcal{L}_{seq}$ | 29.5 | 9.8 | 25.5 |
| Seq2Seq+$\mathcal{L}_{seq}$+$\mathcal{L}_{copy}$ | **30.1** | **10.1** | **26.0** |

GNMT translations, on the other hand, ignores or misinterprets some of the key terms. More examples are provided in Appendix E.

We also test the robustness of our method wrt the hyper-parameter $\gamma$. Results are summarized in Appendix C. Our OT-augmented model is robust to the choice of $\gamma$. The test BLEU scores are consistently higher than the baseline for $\gamma \in (0, 1]$.

## 5.2 ABSTRACTIVE TEXT SUMMARIZATION

We consider two datasets for abstractive text summarization. The first one is the Gigaword corpus (Graff et al., 2003), which has around 3.8M training samples, 190K validation samples, and 1951 test samples. The input pairs consist of the first sentence and the headline of an article. We also evaluate our model on the DUC-2004 test set (Over et al., 2007), which consists of 500 news articles. Our implementation of the Seq2Seq model adopts a simple architecture, which consists of a bidirectional GRU encoder and a GRU decoder with attention mechanism (Bahdanau et al., 2015)[3].

Results are summarized in Tables 4 and 5. Our OT-regularized model outperforms respective baselines. The state-of-the-art ROUGE result for the Gigawords dataset is 36.92 reported by Wang et al. (2018a). However, much more complex architectures are used to achieve that score. We use a relatively simple Seq2Seq model in our experiments to demonstrate the versatility of the proposed OT method. Applying it for ($i$) more complicated models and ($ii$) more recent datasets such as CNN/DailyMail (See et al., 2017) will be interesting future work.

Summarization examples are provided in Appendix D. Similar to the machine translation task, our proposed method captures the key semantic information in both the source and reference sentences.

## 5.3 IMAGE CAPTIONING

We also consider an image captioning task using the COCO dataset (Lin et al., 2014), which contains 123,287 images in total and each image is annotated with at least 5 captions. Following Karpathy's

---

[3]`https://github.com/thunlp/TensorFlow-Summarization`

Table 6: Results for image captioning on the COCO dataset.

| Method | BLEU-1 | BLEU-2 | BLEU-3 | BLEU-4 | METEOR | CIDEr |
|---|---|---|---|---|---|---|
| Soft Attention (Xu et al., 2015) | 70.7 | 49.2 | 34.4 | 24.3 | 23.9 | - |
| Hard Attention (Xu et al., 2015) | 71.8 | 50.4 | 35.7 | 25.0 | 23.0 | - |
| Show & Tell (Vinyals et al., 2015) | - | - | - | 27.7 | 23.7 | 85.5 |
| ATT-FCN (You et al., 2016) | 70.9 | 53.7 | 40.2 | 30.4 | 24.3 | - |
| SCN-LSTM (Gan et al., 2017) | 72.8 | 56.6 | 43.3 | 33.0 | 25.7 | 101.2 |
| Adaptive Attention (Lu et al., 2017) | 74.2 | 58.0 | 43.9 | 33.2 | 26.6 | 108.5 |
| Top-Down Attention (Anderson et al., 2018) | 77.2 | — | — | 36.2 | 27.0 | 113.5 |
| **No attention, Resnet-152** | | | | | | |
| Show & Tell | 70.3 | 53.7 | 39.9 | 29.5 | 23.6 | 87.1 |
| *Show & Tell+$\mathcal{L}_{seq}$ (Ours)* | **70.9** | **54.2** | **40.4** | **30.1** | **23.9** | **90.0** |
| **No attention, Tag** | | | | | | |
| Show & Tell | 72.1 | 55.2 | 41.3 | 30.1 | 24.5 | 93.4 |
| *Show & Tell+$\mathcal{L}_{seq}$ (Ours)* | **72.3** | **55.4** | **41.5** | **31.0** | **24.6** | **94.7** |
| **Soft attention, FastRCNN** | | | | | | |
| Show, Attend & Tell | 74.0 | 58.0 | 44.0 | 33.1 | 25.2 | 99.1 |
| *Show, Attend & Tell+$\mathcal{L}_{seq}$ (Ours)* | **74.5** | **58.4** | **44.5** | **33.8** | **25.6** | **102.9** |

split (Karpathy & Fei-Fei, 2015), 113,287 images are used for training and 5,000 images are used for validation and testing. We follow the implementation of the Show, Attend (Xu et al., 2015)[4], and use Resnet-152 (He et al., 2016), image tagging (Gan et al., 2017), and FastRCNN (Anderson et al., 2018) as the image feature extractor (encoder), and a one-layer LSTM with 1024 units as the decoder. The word embedding dimension is set to 512. Note that in this task, the input are images instead of sequences, therefore $\mathcal{L}_{copy}$ cannot be applied.

We report BLEU-$k$ ($k$ from 1 to 4) (Papineni et al., 2002), CIDEr (Vedantam et al., 2015), and METEOR (Banerjee & Lavie, 2005) scores and the results with different settings are shown in Table 6. Consistent across-the-board improvements are observed with the introduction of the OT loss, in contrast to the RL-based methods where drastic improvements can only be observed for the optimized evaluation metric (Rennie et al., 2017). Consequently, the OT loss is a more reliable method to improve the quality of generated captions when compared with RL methods that aim to optimize and therefore potentially overfit one specific metric. Examples of generated captions are provided in Appendix F.

## 6 CONCLUSION

This work is motivated by the major deficiency in training Seq2Seq models: that the MLE training loss does not operate at sequence-level. Inspired by soft bipartite matching, we propose the usage of optimal transport as a sequence-level loss to improve Seq2Seq learning. By applying this new method to machine translation, text summarization, and image captioning, we demonstrate that our proposed model can be used to help improve the performance compared to strong baselines. We believe the proposed method is a general framework, and will be useful to other sequence generation tasks as well, such as conversational response generation (Li et al., 2017; Zhang et al., 2018c), which is left as future work.

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

## APPENDIX

## A  TRAINING DETAILS FOR NMT TASK

The following training details are basically the same as the intructions from the Tensorflow/nmt github repository:

EN-VI  2-layer LSTMs of 512 units with bidirectional encoder (*i.e.*, 1 bidirectional layer for the encoder), embedding dim is 512. LuongAttention (Luong et al., 2015a) (scale=True) is used together with dropout keep-prob=0.8. All parameters are uniformly initialized. We use SGD with learning rate 1.0 as follows: train for 12K steps (around 12 epochs); after 8K steps, we start halving learning rate every 1K step.

DE-EN  The training hyperparameters are similar to the EN-VI experiments except for the following details. The data is split into subword units using BPE (32K operations). We train 4-layer LSTMs of 1024 units with bidirectional encoder (*i.e.*, 2 bidirectional layers for the encoder), embedding dimension is 1024. We train for 350K steps (around 10 epochs); after 170K steps, we start halving learning rate every 17K step.

## B  END-TO-END NEURAL MACHINE TRANSLATION

Table 7: BLEU scores on VI-EN and EN-VI.

| Systems | NT2012 | NT2013 |
|---|---|---|
| VI-EN: GNMT | 20.7 | 23.8 |
| VI-EN: *GNMT+OT (Ours)* | 21.9 | 25.5 |
| EN-VI: GNMT | 23.0 | 25.4 |
| EN-VI: *GNMT+OT (Ours)* | 24.1 | 26.5 |

Table 8: BLEU scores on DE-EN and EN-DE.

| Systems | NT2013 | NT2015 |
|---|---|---|
| DE-EN: GNMT | 28.5 | 29.0 |
| DE-EN: *GNMT+OT (Ours)* | 28.8 | 29.5 |
| EN-DE: GNMT | 23.7 | 25.3 |
| EN-DE: *GNMT+OT (Ours)* | 24.1 | 26.2 |

Table 7 and 8 show the quantitative comparison for training from random initialization.

## C    BLEU SCORE FOR DIFFERENT HYPER-PARAMETERS

We tested different $\gamma$ for the OT loss term and summarized the results in Figure 3. $\gamma = 0.1$ gave the best performance for the EN-VI experiment. The results are robust wrt the choice of $\gamma \leq 1.0$.

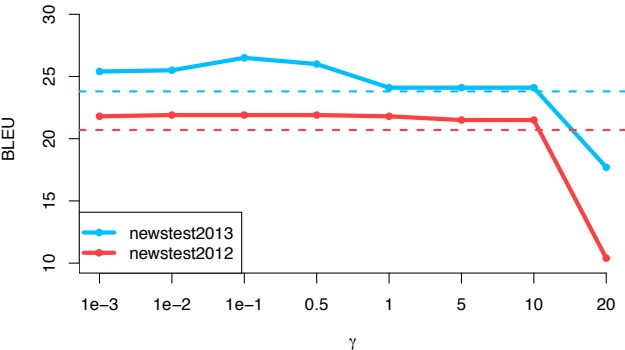

Figure 3: the performance of EN-VI translation by different $\gamma$.

## D    SUMMARIZATION EXAMPLES

Table 9: Examples on Text Summarization.

| Examples | |
| --- | --- |
| Source: | japan 's nec corp. and UNK computer corp. of the united states said wednesday they had agreed to join forces in supercomputer sales . |
| Reference: | nec UNK in computer sales tie-up |
| Baseline: | nec UNK computer corp. |
| Ours: | nec UNK computer corp sales supercomputer. |
| Source: | five east timorese youths who scaled the french embassy 's fence here thursday , left the embassy on their way to portugal friday . |
| Reference: | UNK latest east timorese asylum seekers leave for portugal |
| Baseline: | five east timorese youths leave embassy |
| Ours: | five east timorese seekers leave embassy for portugal |
| Source: | the us space shuttle atlantis separated from the orbiting russian mir space station early saturday , after three days of test runs for life in a future space facility , nasa announced . |
| Reference: | atlantis mir part ways after three-day space collaboration by emmanuel UNK |
| Baseline: | atlantis separate from mir |
| Ours: | atlantis separate from mir space by UNK |
| Source: | australia 's news corp announced monday it was joining brazil 's globo , mexico 's grupo televisa and the us tele-communications inc. in a venture to broadcast ### channels via satellite to latin america . |
| Reference: | news corp globo televisa and tele-communications in satellite venture |
| Baseline: | australia 's news corp joins brazil |
| Ours: | australia 's news corp joins brazil in satellite venture |

Examples for abstract summarization are provided in Table 9.

## E    NEURAL MACHINE TRANSLATION EXAMPLES

In Table 10, we show more examples for comparison. From these examples, sentences generated from our model are more faithful to the reference sentences.

## F    IMAGE CAPTION EXAMPLES

Table 4 shows the comparison of our model with other baselines.

## G    ENCODING MODEL BELIEF WITH SOFTMAX AND GUMBEL-SOFTMAX

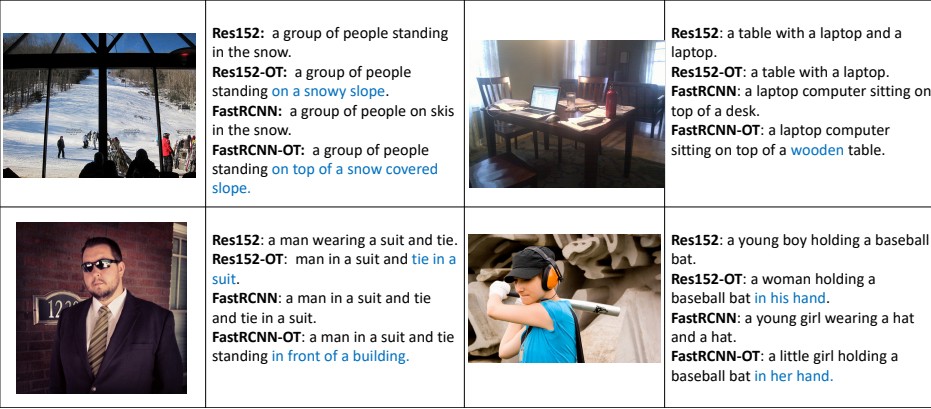

Figure 4: Examples of image captioning on MS COCO.

To find out the best differentiable sequence generating mechanism, we also experimented with Softmax and Gumbel-softmax (a.k.a. concrete distribution). Detailed results are summarized in Table 11. We can see Softmax and Gumbel-softmax based OT model provide less significant gains in terms of BLEU score compared with the baseline MLE model. In some situation, the performance even degenerate. We hypothesized that this is because Softmax encodes more ambiguity and Gumbel-softmax has a larger variance due to the extra random variable involved. These in turn hurts the learning. More involved variance reduction scheme might offset such negative impacts for Gumbel-softmax, which is left as our future work.

Table 11: BLEU scores on VI-EN and EN-VI using different choices of model's belief.

| Systems | NT2012 | NT2013 |
|---|---|---|
| VI-EN: GNMT | 20.7 | 23.8 |
| VI-EN: *GNMT+OT (GS)* | 20.9 | 24.5 |
| VI-EN: *GNMT+OT (softmax)* | 21.8 | 24.3 |
| VI-EN: *GNMT+OT (ours)* | **21.9** | **25.5** |
| EN-VI: GNMT | 23.0 | 25.4 |
| EN-VI: *GNMT+OT (GS)* | 23.3 | 25.7 |
| EN-VI: *GNMT+OT (softmax)* | 23.5 | 26.0 |
| EN-VI: *GNMT+OT (ours)* | **24.1** | **26.5** |

## H  OT IMPROVES BOTH MODEL AND WORD EMBEDDING MATRIX

To identify the source of performance gains, we designed a toy sequence-to-sequence experiment to show that OT help to refine the language model and word embedding matrix. We use the English corpus from WMT dataset (from our machine translation task) and trained an auto-encoder (Seq2Seq model) on this dataset. We evaluated the reconstruction quality with the BLEU score. In Case 1, we stop the OT gradient from flowing back to the sequence model ( only affecting the word embedding matrix); while in Case 2, the gradient from OT can affect the entire model. Detailed results are shown in Table G. We can see that Case 1 is better than the baseline model, which means OT helps to refine the word embedding matrix. Case 2 achieves the highest BLEU, which implies OT also helps to improve the language model.

Table 12: Comparison experiment.

| Metric | Baseline | Case 1 | Case 2 |
|---|---|---|---|
| BLEU-2 | 71.87 | 73.60 | **75.12** |
| BLEU-3 | 61.18 | 63.07 | **64.82** |
| BLEU-4 | 56.59 | 58.48 | **60.27** |
| BLEU-5 | 53.73 | 55.69 | **57.50** |

Table 10: More DE-EN translation examples.

| Examples | |
|---|---|
| Reference: | When former First Lady Eleanor Roosevelt chaired the International Commission on Human Rights, which drafted the Universal Declaration of Human Rights that would in 1948 be adopted by the United Nations as a global covenant, Roosevelt and the drafters included a guarantee that "everyone has the right to form and to join trade unions for the protection of his interests." |
| Ours: | When former First Lady Eleanor Roosevelt held the presidency of the International Commission on Human Rights , drafted by the Universal Declaration of Human Rights , adopted in 1948 by the United Nations as a global agreement , Roosevelt and the other authors gave a guarantee that " everyone has the right to form or join trade unions to protect their interests . " |
| GNMT: | When former First Lady Eleanor Roosevelt presided over the International Human Rights Committee , which drew up the Universal Declaration of Human Rights , as adopted by the United Nations in 1948 as a global agreement , Roosevelt and the other authors added a guarantee that " everyone has the right to form trade unions to protect their interests or to accede to them " . |
| Reference: | India's new prime minister, Narendra Modi, is meeting his Japanese counterpart, Shinzo Abe, in Tokyo to discuss economic and security ties, on his first major foreign visit since winning May's election. |
| Ours: | India s new Prime Minister Narendra Modi meets his Japanese counterpart , Shinzo Abe , in Tokyo at his first major foreign visit since his election in May in order to discuss economic and security relations . |
| GNMT: | India s new prime minister , Narendra Modi , is meeting his Japanese counterpart , Shinzo Abe , in Tokyo at his first important visit abroad in May to discuss economic and security relations . |
| Reference: | The police used tear gas . |
| Ours: | The police used tear gas . |
| GNMT: | The police put in tear gas . |
| Reference: | There were three people killed. |
| Ours: | Three people were killed . |
| GNMT: | Three people had been killed . |
| Reference: | The next day, turning up for work as usual, she was knocked down by a motorcyclist who had mounted the pavement in what passers-by described as a "vicious rage." |
| Ours: | The next day , when she went to work as usual , she was crossed by a motorcyclist who , as described by passers-by , was in a sort of " brutal rage " on the road . |
| GNMT: | The next day , when she went to work as usual , she was driven by a motorcyclist who , as passants described , went on foot in a kind of " brutal anger " . |
| Reference: | Double-check your gear. |
| Ours: | Check your equipment twice . |
| GNMT: | Control your equipment twice . |
| Reference: | Chinese leaders presented the Sunday ruling as a democratic breakthrough because it gives Hong Kongers a direct vote, but the decision also makes clear that Chinese leaders would retain a firm hold on the process through a nominating committee tightly controlled by Beijing. |
| Ours: | The Chinese leadership presented Sunday s decision as a democratic breakthrough because it gives the citizens of Hong Kong a direct right to vote , but the decision also makes it clear that the Chinese leadership keeps the process firmly in the hands of a government-controlled Nomination Committee . |
| GNMT: | The Chinese leadership presented Sunday s decision as a democratic breakthrough , because Hong Kong s citizens have a direct right to vote , but the decision also makes it clear that the Chinese leadership is firmly in control of the process through a UN-controlled nomination committee . |
| Reference: | Her mother arrived at Mount Sinai Hospital Thursday after an emergency call that she was in cardiac arrest at an Upper East Side clinic, Yorkville Endoscopy, sources said. |
| Ours: | According to sources , her mother was sent to Mount Sinai on Thursday after an emergency due to heart failure in a clinic at the Upper East Side , Yorkville Endoscopy . |
| GNMT: | According to sources , her mother was sent to Mount Sinai hospital on Thursday after an emergency due to heart closure in a clinic at the Upper East Side , Yorkville Endoscopy . |
| Reference: | Ukrainian soldiers had to withdraw from their positions in Ilovaysk after two columns of Russian armor and 1,000 troops last week moved into the Donetsk region to bolster the beleaguered separatists, Col. Andriy Lysenko, spokesman for the Ukrainian National Security and Defense Council, told reporters in Kiev on Saturday. |
| Ours: | Ukrainian soldiers had to withdraw from their positions in Ilowajsk after two Russian tanks and 1,000 soldiers entered the Donetsk region last week to support the belated separatists , said Colonel Andrij Lysenko , Speaker of the Ukrainian National Security and Defence Council Reporters on Saturday in Kiev . |
| GNMT: | Ukrainian soldiers had to withdraw from their positions in Ilowajsk after two Russian tanks and 1,000 soldiers invaded the Donetsk region last week to support the beloved separatists , said Colonel Andriy Lysenko , Ukrainian National Security and Defence Council spokesman on Saturday in Kiev . |
| Reference: | Mountain Rescue doctor, Professor Volker Lischke, who was there with his team to provide safety, and who was equipped with a four-wheel Bully and Quad, said: "I know him from Frankfurt - he trains for a specialist sleigh trail - it's just that he pulls the sleigh himself." The man is, therefore, in a sense his own sleigh dog. |
| Ours: | Bergwacht doctor Professor Volker Lischke , who , with his team , endowed with Allrad-Bully and Quad for safety , said : " The kenn " I from Frankfurt , trained for a special sleigh trail , just that he trains the sleigh himself " , so the man is , in a sense , his own sled dog . |
| GNMT: | Bergwacht physician Professor Volker Lischke , who with his team , equipped with Allrad-Bully and Quad , for safety , said : " Den kenn " ich aus Frankfurt , which trained for a special Schlittentrail , only to stop the sleigh itself " , so the man is in some way his own hook dog . |

