# OpenReview forum: "Improving Sequence-to-Sequence Learning via Optimal Transport"
_ICLR.cc/2019/Conference_

### Official Review · AnonReviewer2 · 2018-10-31
**An OT-based regularization of the loss of seq2seq models**

**Rating:** 5
**Confidence:** 4

**Review:**

This paper propose to add an OT-based regularization term to seq-2-seq models in order to better take into account the distance between the generated and the reference and/or source sentences, allowing one to capture the semantic meaning of the sequences. Indeed, it allows the computation of a distance between embeddings of a set of words, and this distance is then used to define a penalized objective function.
The main issue with this computation is that it provides a distance between a set of words but not a sequence of words. The ordering is then not taken into account. Authors should discuss this point in the paper.
Experiments show an improvement of the method w.r.t. not penalized loss.

Minor comments:
- in Figure 1, the OT matching as described in the text is not the solution of eq (2) but rather the solution of eq. (3) or the entropic regularization (the set of "edges" is higher than the admissible highest number of edges).
- Introduction "OT [...] providing a natural measure of distance for sequences comparisons": it is not clear why this statement is true. OT allows comparing distributions, with no notion of ordering (see above).
- Table 1: what is NMT?
- first paragraph, p7: how do you define a "substantial" improvement of the scores?
- how do you set parameter $\gamma$ in the experiments? Why did you choose \beta=0.5 for the ipot algorithm?

---

> ### Author Response · Authors · 2018-11-14
> **Authors' response**
>
> Dear AnonReviewer2:
>
> Thank you very much for your comments!
> You can find our point-to-point response to your concerns below:
>
> 1. OT only considers a set of words not a sequence of word:
> This is a valid point, and it is the exact reason why we have also used MLE loss to enforce the order consistency (which referred to as the syntax part in the manuscript). The OT regularization we introduced aims to improve the semantic consistency during model training.
>
> We believe further applying optimal transport can improve the matching of the semantic contents (e.g., key words) between the sequences, so as to help the model match key words and maintain the semantic meaning. Theoretical justification can be found in Section 3.
>
> The optimal transport can also be perceived as a soft-copying mechanism to preserve the key content from source to target. This is related to the CopyNet model [C1], which does not consider the ordering information either. Detailed discussion can be found in Section 2.2, paragraph "Soft-copying mechanism".
>
> A more detailed discussion has been added to our paper to clarify this (last paragraph of Section 2 and the first paragraph of Section 4). We also have added an experiment to show that our method can improve both the language model and word embedding matrix, details can be found in Appendix H.
>
> Minor points:
>
> 1. Some confusions on Figure 1 and OT defined in Eqn (2) \& (3):
> To clarify, Eqn (3) is the numerical scheme used to solve the problem defined in Eqn (2). As such, they are equivalent. Sinkhorn differs slightly because it solves a entropy regularized OT problem, which we do not use. Additionally, in Eqn (2) we did not set the "admissible highest number of edges". We think this may be a hard constraint and is not clear how to optimize it in practice.
>
> 2. Why is OT a natural measure for sequence comparisons, as no ordering information is involved: Admittedly, our OT objective does not explicitly consider ordering information. We basically follow the argument from Word mover's distance (WMD) [C3] to consider a bag-of-words similarity measures, where WMD demonstrates great success in measuring text similarity. We considered this to be ``natural'' because our measure is based on similarity in embedding space, rather than hard-matching. We believe the order information could be helpful. However how to leverage such ordering information in OT objective design in an efficient manner is not trivial  and can be an interesting future work.
>
> 3. What is NMT in Table 1:
> Thanks for pointing out this issue. In Table 1, NMT refers to Google's Neural Machine Translation model. We have made "GNMT" and "NMT" consistent in Table 1 in our revision.
>
> 4. How do you define "substantial" improvement of the scores:  We agree that the phrase "substantial" can be subjective. We rephrase it to ``consistent improvements''.
>
> 5. Choice of hyper-parameters:
> We set $\gamma=0.1$ in our experiments, Figure 3 in the Appendix C, shows how the performance changes with respect to different values of $\gamma$. $\beta$ is the parameter for the proximal point methods, which is fairly robust for different values [C1] and only affects the convergence rate. We choose $\beta=0.5$, since it helps converge faster than other setups. We observed that as long as $\beta$ is within a reasonable range, i.e., (0,10], the results are not sensitive to this hyperparameter.
>
> [C1] Xie, et.al (2018), A Fast Proximal Point Method for Computing Wasserstein Distance
> [C2] Gu et al., Incorporating copying mechanism insequence-to-sequence learning. ACL 2016.
> [C3] Kusner et al., From Word Embeddings To Document Distances. ICML 2015.

---

### Official Review · AnonReviewer1 · 2018-11-02
**Sequence level regularization based on optimal transport**

**Rating:** 7
**Confidence:** 4

**Review:**

This submission deals with text generation. It proposes a sequence level objective, motivated by optimal transport, which is used as a regularizer to the (more standard) MLE. The goal is to complement MLE by allowing `soft matches` between different tokens with similar meanings. Empirical evaluation shows that the proposed technique improves over baselines on many sequence prediction tasks. I found this paper well motivated and a nice read. I would vote for acceptance.

Pros:
- A well-motivated and interesting method.
- Solid empirical results.
- Writing is clear.

Cons:
- The split of `syntax--MLE` and `semantics--OT` seems a bit awkward to me. Matching the exact tokens does not appear syntax to me.
- Some technical details need to be clarified.

Details:
- Could the authors comment on the efficiency by using the IPOT approximate algorithm, e.g., how much speed-up can one get? I'm not familiar with this algorithm, but is there convergence guarantee or one has to set some kind of maximum iterations when applying it in this model?

- Bottom of page 4, the `Complementary MLE loss paragraph`. I thought the OT loss is used as a regularizer, from the introduction. If the paper claims MLE is actually used as the complements, evidence showing that the OT loss works reasonably well on its own without including log loss, which I think is not included in the experiments.

- I really like the analysis presented in Section 3. But it's a bit hard for me to follow, and additional clarification might be needed.

- It would be interesting to see whether the `soft-copying` version of OT-loss can be combined with the copy mechanisms based on attention weights.

================================

Thanks for the clarification and revision! It addressed some of my concerns. I would stick to the current rating, and vote for an acceptance.

---

> ### Author Response · Authors · 2018-11-14
> **Authors' response**
>
> Dear AnonReviewer1:
>
> Thank you very much for your inputs!
> Below you can find our detailed response to your comments:
>
> 1. Please clarify the algorithmic efficiency and convergence guarantees of the IPOT algorithm:
> The IPOT has the same algorithmic complexity compared with Sinkhorn, but it solves the exact OT problem, not the entropy regularized OT. So it is not necessarily faster, but definitely more accurate. The convergence guarantees of IPOT can be find in Theorem 3.1 from [B1].
>
> 2. Is OT loss a regularizer or main loss:
> The OT is used as a regularizer rather than the main loss. OT training considered in this paper cannot train a proper language model on its own because it does not explicitly consider word ordering. We have fixed our paper in Section 2 correspondingly to reduce the confusion.
>
> 3. More clarifications on the theoretical justification:
> We show in Section 3 that our training objective can be well approximated by a Wasserstein gradient flow, whose solution is the data distribution.
> The key formulation is Equation (6), which is the solution of a WGF. We show that under certain conditions, Equation (6) recovers our training objective. Thus we have built a connection between WGF and our proposed method. We appreciate the reviewer's suggestion and have edited Section 3 to make it easier for readers less familiar with the WGF theory to follow.
>
> 4. It will be good to see additional results on soft-copying OT with attention weights:
> Thanks, this is a very good idea. We will try to pursue this in our future work.
>
>
> [B1] Xie, et.al (2018), A Fast Proximal Point Method for Computing Wasserstein Distance

---

> ### Author Response · Authors · 2018-12-17
> **Thanks for the updated review**
>
> Dear AnonReviewer1:
>
> Thanks very much for the updated review and your valuable time.
>
> Best,
> Authors

---

### Official Review · AnonReviewer3 · 2018-11-05
**Updated score; final comments**

**Rating:** 6
**Confidence:** 3

**Review:**

====== Final Comments =======
I thank the authors for updating the manuscript with clarifications and for clear replies to my concerns.

I agree with R2 to some extent that the empirical performance of the method, as well as the formulation,  is interesting. In general, the authors addressed my concerns regarding how optimal transport training model interfaces with MLE training and the choice of using scaled softmax for computing the Wasserstein distances. However, I still find myself agreeing with R3 that the choice of sets to compute Wasserstein distances (as opposed to sequences is somewhat unjustified); and it is not clear how the theory in Sec. 3 justifies using sets instead of words, as the data distribution p_d is over sequences in both the W-2 term as well as the MLE term. This would be good to clarify further, or explain more clearly how Sec. 3 justifies this choice.

Also, I missed this in the original review, but the assumption that KL(\mu| p_d) = KL(\p_d| \mu) since \mu is close to p_d for MLE training does not seem like a sensible assumption under model misspecification (as MLE is mode covering). I would suggest the authors discuss this in a revision/ camera-ready version.

In light of these considerations, I am updating the rating to 6 (to reflect the points that have been addressed), but I still do not believe that the method is super-well justified, despite an interesting formulation and strong empirical results (which are aspects the paper could still improve upon).
=====================

**Summary**

The paper proposes a regularization / fine-tuning scheme in addition to maximum likelihood sequence training using optimal transport. The basic idea is to match a sampled sentence from a model with a ground truth sentence using optimal transport. Experimental results show that the proposed modification improves over MLE training across machine translation, summarization and image captioning domains.

**Strengths**
+ The proposed approach shows promising results across multiple domains.
+ The idea of using optimal transport to match semantics of words intuitively makes sense.

**Weaknesses**
1. Generally, one would think that for optimal transport we would use probabilities which come from the model, i.e. given a set of words and probabilities for each of the words, one would move mass around from one word to another word (present in the ground truth) if the words were semantically relevant to each other and vice versa. However, it seems the proposed algorithm / model does not get the marginal distributions for the words from the model, and indeed does not use any beliefs from the model in the optimal transport formulation / algorithm [Alg. 1, Line 2]. This essentially then means, following objective 2, that optimal transport has no effect on the model’s beliefs on the correct word, but merely needs to ensure that the cosine similarity between words which have a high transport mass is minimized, and has no direct relation to the belief of the model itself. This strikes as a somewhat odd design choice. (*)

2. In general, another issue with the implementation as described in the paper could be the choice of directly using the temperature scaled softmax as an estimate of the argmax from the model, instead of sampling utterances from the model. It seems worth reporting results, even if as a baseline what sampling from something like a concrete distribution [A] yeilds in terms of results. (*)

3. As a confirmation, Is the differentiable formulation for sentences also used for MLE training part of the objective, or does MLE still use ground truth sentences (one-hot encoded) as input. Does this mean that the model is forwarded twice (once with ground truth, and once with softmax outputs)? In general, instead of/ in addition to an Algorithm box that expains the approach of Xie et.al., it would be good to have a clear algorithmic explanation of the training procedure of the proposed model itself. Further, the paper is not clear if the model is always used for finetuning the an MLE model (as Section 3) seems to suggest or if the optimal transport loss is used in conjunction with the model as the bullet ``complementary MLE loss’’ seems to suggest. (*)

4. Section 3: It is not clear that the results / justifications hold for the proposed model since the distance optimized in the current paper is not a wasserstein-2 distance. Sure, it computes cosine distance, which is L-2 but it appears wasserstein-2 distance is defined for continuous probability measures, while the space on which the current paper computes distances is inherently the (discrete) space of word choices. (*)

Experiments
5. The SPICE metric for image captioning takes the content of the sentences (and semantics) into account, instead of checking for fluency. Prior work on using RL techniques for sequence predictions have used SPICE + CIDEr [B] to alleviate the problem with RL methods mentioned in page. 5. Would be good to weaken the claim. (*)

Minor Points
1. It might be good to be more clear about the objective from Xie et.al. by also stating the modified formulation they discuss, and explaining what choice of the function yeilds the bregman divergence of the form discussed in Eqn. 3.
2. It would be nice to be consistent with how \mathcal{L}_{ot} is used in the paper. Eqn. 4 lists it with two embedding matrices as inputs, while Alg. 1, Line 11 assigns it to be the inner product of two matrices.

**Preliminary Evaluation**
In general, the paper is an interesting take on improving MLE for sequence models. However, while the idea of using optimal transport is interesting and novel for training sequence models, I have questions about the particular way in which it has been implemented, which seems somewhat unjustified. I also have further clarifications about a claimed justification for the model. Given convincing responses for these, and other clarification questions (marked with a *) this would be a good paper.

References
[A]: Maddison, Chris J., Andriy Mnih, and Yee Whye Teh. 2016. “The Concrete Distribution: A Continuous Relaxation of Discrete Random Variables.” arXiv [cs.LG]. arXiv. http://arxiv.org/abs/1611.00712.
[B]: Liu, Siqi, Zhenhai Zhu, Ning Ye, Sergio Guadarrama, and Kevin Murphy. 2016. “Improved Image Captioning via Policy Gradient Optimization of SPIDEr.” arXiv [cs.CV]. arXiv. http://arxiv.org/abs/1612.00370.

---

> ### Author Response · Authors · 2018-11-14
> **Authors' response (1/2)**
>
> Dear AnonReviewer3:
>
> Thank you very much for your comments!
> Your comments have been carefully addressed below:
>
> 1. How does the OT objective leverage the model's belief:
> We believe reviewer's confusion can be attributed to the fact that the notations we employed for the IPOT algorithm is no entirely consistent with the rest of our paper (which the reviewer also kindly pointed out as a minor issue). To clarify, our model does take the model's belief into account.
> While the marginal distribution of the words is not directly fed into the IPOT algorithm, it is used to compute the model's predicted embedding. As such, both the model's belief and word embedding are updated when training with the OT objective. More specifically, we have $z_i = E^T w_i, z'_i = E^T \hat{w}_i$ for feature vectors $\{S_i\}$ and $\{ S'_i \}$ used in Alg 1 as input, where $E$ denotes the word embedding matrix, $\{w_i\}$ is the one-hot encoding vector for the $i$-th word in the sentence and $\hat{w}_i$ is the model's belief for the word at that location. This means the model's belief is encoded in $\{z'_i\}$. Alg 1 Ln 2 is just the parameter initialization step for the IPOT algorithm, which of course has zero information on the model's belief. The T matrix will assimilate the model's belief from $\{z'_i\}$ as the IPOT algorithm starts to iterate. We also remark there are a number of different ways to encode the model's belief.  In this study, we have experimented with several different popular choices (Average, SoftArgmax, Gumbel/Concrete, etc.) to identify which works best. It turned out that the Soft-argmax approach yields the best empirical performance for the evaluation metrics we considered. In our original submission we did not make this clear enough due to space limitations. In response to the reviewer's comment, we have added more discussions on this point to the paper and included more experiment details in the Appendix G and H, in Page 15. The notation inconsistency has been fixed as well.
>
> 2. About the choice of SoftArgmax rather than the recently proposed Concrete estimator (a.k.a. Gumbel Softmax (GS)):
> We agree with the reviewer that alternative estimators should be discussed. We did consider applying GS in our experiments and found it renders less-stable training, possibly due to a higher variance compared with temperature-annealed Soft-argmax. In our experiments, using GS often give us sub-optimal solution, and at times even worse than the MLE baseline. Consequently, we only reported results with the Soft-argmax in our original submission. In the revised manuscript, we also report the results of GS estimator in Appendix G, Table 15.
>
> 3. Clarification on the overall training procedure, especially for the MLE part:
> Thanks for the comment. We confirm that the MLE part of our model is the standard MLE training which the uses one-hot encoded ground-truth sentence as input. During training, our model is forwarded once. The algorithm box for training the entire model can be found as Algorithm 2 in Section 2 (Page 5). The OT uses the (annealed) Softmax (soft-argmax) weighted embedding, rather than the embedding of sampled word, at each location to avoid excessive variance. OT loss complements MLE loss and should not be used alone. We have demonstrated its utility both in the fine-tuning stage (after pre-training, as justified in Sec 3) and also right from the beginning of training. Details can be found in Appendix B, Table 7 and 8. We have made these points more clear in the experiment section.
>
> 4. There seems to be a small gap between the theory from Sec 3 and the practice:
> We thank the reviewer for pointing out this confusion, which we hope to resolve as follows. First, Wasserstein distance can be defined for both continuous and discrete distributions [A1], so we believe the reviewer's concern is about the theory regarding Wasserstein gradient flows (WGF). While we have only described the theory for the continuous case (for the sake of simplicity), it actually also holds for the discrete case (see [A2]). These have been clarified in the updated manuscript.
>
> 5. Some prior work has tried to address the weakness of RL we criticized:
> Thanks for bringing this paper to our attention. In our paper, we narrow down our consideration to a single metric as reward. In response to the reviewer's comment, we will discuss this literature in our manuscript and rephrase our claims accordingly.
>
> [A1] G, Luise et.al. (2018), Differential Properties of Sinkhorn Approximation for Learning with Wasserstein Distance
> [A2] Li \& Montufar (2018), Natural gradient via optimal transport.

---

> > ### Author Response · Authors · 2018-11-14
> > **Authors' response (2/2)**
> >
> > Minor points:
> > 1. Please be more clear about the Bregman-based algorithm:
> > To clarify, the IPOT algorithm we adopted solves the exact OT problem, not the entropy regularized variant as in Sinkhorn. It is a case of proximal descent scheme, using Bregman divergence as the proximity metric.
> > Sinkhorn regularize the entropy of the solution, while IPOT regularize the Bregman divergence between current and last iterate. These points have been clarified in our revision.
> >
> > 2. Inconsistent notation of the OT loss:
> > Thanks for pointing this out and we have fixed this issue.

---

> ### Author Response · Authors · 2018-12-17
> **Thanks for the updated review**
>
> Dear AnonReviewer3:
>
> Thanks for your updated comments, we will continue revising our draft to make sure our method is well-justified.
>
> Thanks again for your valuable time.
>
> Best,
> Authors

---

### Author Response · Authors · 2018-11-14
**Thanks for all these insightful comments!**

We would like to thank all the reviewers for taking their time to contribute these insightful comments, which helped us to improve the original submission. Our detailed point-to-point response can be found in our individual replies to the reviews, and we have also carefully updated the manuscript following the constructive suggestions from the reviewers.

Here is a brief summary of major updates made to the manuscript:
1. Clarifications on the IPOT algorithm (Sec 2.1).
2. Discussions on alternative model belief encoding schemes such as Gumbel-softmax, further experiment results updated to the Appendix. (Sec 2.2)
3. Additional experiments showing the proposed OT-regularization can improve both word embedding matrix and language model.
4. Section 3 has been edited to make it easier to follow for readers less familiar with the theory of Wasserstein gradient flows.
5. A new algorithm block describing our full training procedure. (pp. 5)
6. Updated notation system to reduce confusion.

---

### Meta-Review · Area_Chair1 · 2018-12-14
**Accept**

**Confidence:** 4
**Recommendation:** Accept (Poster)

**Metareview:**

The paper proposes the idea of using optimal transport to evaluate the semantic correspondence between two sets of words predicted by the model and ground truth sequences. Strong empirical results are presented which support the use of optimal transport in conjunction with log-likelihood for training sequence models. I appreciate the improvements to the manuscript during the review process, and I encourage the authors to address the rest of the comments in the final version.